# Stable antibiotic resistance and rapid human adaptation in livestock-associated MRSA

**Marta Matuszewska†, Gemma GR Murray*†, Xiaoliang Ba, Rhiannon Wood, Mark A Holmes, Lucy A Weinert**

Department of Veterinary Medicine, University of Cambridge, Cambridge, United Kingdom

**Abstract** Mobile genetic elements (MGEs) are agents of horizontal gene transfer in bacteria, but can also be vertically inherited by daughter cells. Establishing the dynamics that led to contemporary patterns of MGEs in bacterial genomes is central to predicting the emergence and evolution of novel and resistant pathogens. Methicillin-resistant *Staphylococcus aureus* (MRSA) clonal-complex (CC) 398 is the dominant MRSA in European livestock and a growing cause of human infections. Previous studies have identified three categories of MGEs whose presence or absence distinguishes livestock-associated CC398 from a closely related and less antibiotic-resistant human-associated population. Here, we fully characterise the evolutionary dynamics of these MGEs using a collection of 1180 CC398 genomes, sampled from livestock and humans, over 27 years. We find that the emergence of livestock-associated CC398 coincided with the acquisition of a Tn*916* transposon carrying a tetracycline resistance gene, which has been stably inherited for 57 years. This was followed by the acquisition of a type V SCC*mec* that carries methicillin, tetracycline, and heavy metal resistance genes, which has been maintained for 35 years, with occasional truncations and replacements with type IV SCC*mec*. In contrast, a class of prophages that carry a human immune evasion gene cluster and that are largely absent from livestock-associated CC398 have been repeatedly gained and lost in both human- and livestock-associated CC398. These contrasting dynamics mean that when livestock-associated MRSA is transmitted to humans, adaptation to the human host outpaces loss of antibiotic resistance. In addition, the stable inheritance of resistance-associated MGEs suggests that the impact of ongoing reductions in antibiotic and zinc oxide use in European farms on livestock-associated MRSA will be slow to be realised.

**\*For correspondence:**
ggrmurray@gmail.com

†These authors contributed equally to this work

**Competing interest:** The authors declare that no competing interests exist.

## Editor's evaluation

The reviewers recognised the importance of understanding where new strains of microbes come from and how they change over time for infection control and prevention. *Staphylococcus aureus* CC398 is an important strain that 'spills over' from livestock to humans, carrying with it high levels of resistance to antibiotics commonly used in farming. This paper compares more than 1000 genomes of CC398 and concludes that spillover is likely to carry resistance to tetracyclines and other antibiotics into humans that will persist over time.

## Introduction

Mobile genetic elements (MGEs) play an important role in the evolution of bacterial pathogens. They can move rapidly between bacterial genomes, but can also be vertically inherited through stable integration into a host genome. As MGEs often carry genes associated with virulence and antibiotic

**eLife digest** Antibiotic-resistant infections are a growing threat to human health. In 2019, these hard-to-treat infections resulted in 4.95 million deaths making them the third leading cause of death that year. Excessive use of antibiotics in humans is likely driving the emergence of drug-resistant bacteria. But there is a concern that use of antibiotics on livestock farms is also contributing. A type of bacteria traced back to livestock is a growing cause of human infections that do not respond to treatment with the antibiotic methicillin in Europe. It is called livestock-associated methicillin-resistant *Staphylococcus aureus* (LA-MRSA).

Bacteria can share genes that make them drug resistant or more deadly. These genes are often carried on mobile genetic elements that promote their movement from one bacterial cell to another. The most common type of LA-MRSA in Europe is clonal-complex 398 (CC398). It has two mobile genetic elements carrying antibiotic-resistance genes, but generally lacks a mobile genetic element that helps the bacterium escape the human immune system. Learning more about how LA-MRSA acquired these genetic changes may help scientists develop better strategies to protect the public.

Matuszewska, Murray et al. analyzed the genomes of more than 1,000 samples of CC398 collected from humans, pigs and 13 other animal species in 28 countries over 27 years. They used this data to reconstruct the bacteria's evolutionary history. Matuszewska, Murray et al. show that two mobile elements containing antibiotic resistance genes in CC398 were gained decades ago. One is more than 50 years old and was likely acquired around the time antibiotic use in livestock became common. While most CC398 in livestock do not have a mobile element that helps LA-MRSA evade the human immune system, they often gain it when they infect humans. This leads to highly drug-resistant human MRSA infections.

The results of this study suggest that LA-MRSA is a serious threat to human health. The resistance of this bacterium has persisted for decades, spreading across different livestock species and different countries. These drug-resistant bacteria in livestock readily infect humans. Current efforts to reduce antibiotic use in farms may take decades to mitigate these risks. Additionally, the ban on zinc-oxide use on livestock in the European Union (coming into force June 2022) may not help reduce LA-MRSA, because the genes conferring resistance to bacteria and zinc treatment are not always linked.

resistance (*Frost et al., 2005*; *Rankin et al., 2011*), an understanding of the drivers and barriers to their acquisition and maintenance is central to predicting the emergence and evolution of novel and resistant pathogens (*Brockhurst et al., 2019*).

The emergence and evolution of methicillin-resistant *Staphylococcus aureus* (MRSA) across different ecological niches and host species are associated with the horizontal transfer of MGEs. Methicillin resistance is carried by the staphylococcal cassette chromosome element SCC*mec*, and additional MGEs carry resistance to other antibiotics, virulence factors, and host-specific adaptations (*Hanssen and Ericson Sollid, 2006*; *Jamrozy et al., 2017*; *Haag et al., 2019*; *Turner et al., 2019*; *Matuszewska et al., 2020*). While most MRSA clonal complexes (CCs) show an association with specific MGEs, their dynamics are not widely understood, leading to a gap in our understanding of the adaptive potential of *S. aureus* CCs.

Intensification combined with high levels of antibiotic use in farming has led to particular concerns about livestock as reservoirs of antibiotic-resistant human infections (*World Health Organization, 2015*). CC398 has become the dominant MRSA in European livestock. Its rise has been particularly evident in Danish pig farms where the proportion of MRSA-positive herds has increased from <5% in 2008 to 90% in 2018 (*Sieber et al., 2018*; *DANMAP, 2019*), but it has also been observed in other European countries, and other livestock species (*Lekkerkerk et al., 2015*; *Islam et al., 2017*; *Anjum et al., 2019*). Livestock-associated (LA) MRSA CC398 has been associated with increasing numbers of human infections, in both people with and without direct contact with livestock (*Larsen et al., 2017*; *van Alen et al., 2017*; *Sieber et al., 2019*). Understanding the emergence and success of CC398 in European livestock and its capacity to infect the human host is integral to managing the risk that it, and other livestock-associated pathogens, pose to public health.

Previous studies have used genome sequences to reconstruct the evolutionary history of CC398 (*Price et al., 2012*; *Ward et al., 2014*; *Gonçalves da Silva et al., 2017*). They identified a largely

methicillin-resistant and livestock-associated clade of CC398, that falls as either sister to *Ward et al., 2014* or within (*Price et al., 2012*; *Gonçalves da Silva et al., 2017*) a largely methicillin-sensitive and human-associated clade. Through comparing the genomes of isolates from livestock- and human-associated CC398, these studies concluded that the emergence of CC398 in livestock was associated with both the acquisition of antibiotic resistance genes and the loss of genes associated with human immune evasion. While these genes are known to be carried on three categories of MGEs (Tn*916* conjugative transposons, SCC*mec*, and φSa3 prophages), little is known about the dynamics of these MGEs within CC398. Here, we undertake a comprehensive reconstruction of the evolutionary dynamics of these MGEs. We find that while their patterns of presence/absence all show a strong association with the transition to livestock, this is the result of contrasting dynamics. These dynamics can inform predictions about the risk posed by LA-MRSA spillover infections in humans, and the resilience of antibiotic resistance in LA-MRSA to ongoing changes in antibiotic use in farming.

## Results

### Livestock-associated CC398 emerged between 1957 and 1970

We collected and assembled publicly available whole-genome sequence data from CC398, and sequenced five isolates recently sampled from pig farms in the United Kingdom. Our collection includes high-quality whole genome assemblies of 1180 isolates (including 43 complete reference genomes). This collection spans 15 host species (including humans, pigs, cows, chickens, turkeys, and horses), 28 countries (across Europe, America, Asia, and Australasia), and 27 years (1992–2018) (*Figure 1—figure supplement 1* and *Supplementary file 1*).

We constructed a recombination-stripped maximum likelihood phylogeny of CC398 using reference-mapped assemblies of our collection. We rooted the phylogeny with outgroups from four other *S. aureus* sequence types (STs) in four separate reconstructions. We also constructed a phylogeny from a recombination-stripped concatenated alignment of core genes extracted from *de novo* assemblies, with a midpoint rooting. These reconstructions consistently returned the same topology, and one that is described in previous studies: a livestock-associated clade of CC398 (704 isolates) that falls within a more diverse, and other largely human-associated clade (476 isolates) (*Figure 1a* and *Figure 1—figure supplement 2*; *Price et al., 2012*; *Gonçalves da Silva et al., 2017*).

We used the temporal structure in our collection to date the origin of the livestock-associated clade. Due to the size of our collection, we constructed dated phylogenies from three subsampled datasets, each of which includes 250 isolates. Our estimates of the evolutionary rate were consistent across all reconstructions (1.1–1.6 × 10⁻⁶ subs/site/year), and similar to estimates from previous studies of CC398 (*Ward et al., 2014*) and other *S. aureus* CCs (*Hsu et al., 2015*). This led to an estimate of the origin of the livestock-associated clade of approximately 1964 (95% CI: 1957–1970) (*Figure 1b*, *Figure 1—figure supplement 3*, and *Figure 1—figure supplement 4*).

### The transition to livestock association is associated with changes in the frequencies of three MGEs with very different dynamics

Comparisons of the genomes of isolates from human- and livestock-associated CC398 have previously indicated that the transition to livestock was associated with the acquisition of genes associated with both tetracycline and methicillin resistance (*tetM* and *mecA*), and the loss of genes associated with human immune evasion (the immune evasion gene cluster) (*Price et al., 2012*). Our analyses of this larger collection are broadly consistent with this. We find that the genes whose presence most strongly distinguishes isolates from the human- and livestock-associated groups are associated with three categories of MGEs: (1) a Tn*916* transposon carrying *tetM*, (2) SCC*mec* carrying *mecA*, and (3) φSa3 prophages carrying a human immune evasion gene cluster (*Figure 1a*, *Figure 1—figure supplement 5*, and *Supplementary file 2*). Genes associated with the Tn*916* transposon and SCC*mec* elements are more common in the livestock-associated clade, while the reverse is true of genes associated with φSa3 prophages.

#### Stable maintenance of a Tn*916* transposon carrying *tetM*

We identified a contiguous assembly of a Tn*916* transposon carrying *tetM* in 699/704 isolates in our collection of livestock-associated CC398 (*Figure 2a*, *Supplementary files 3 and 4*; *de Vries et al.,*

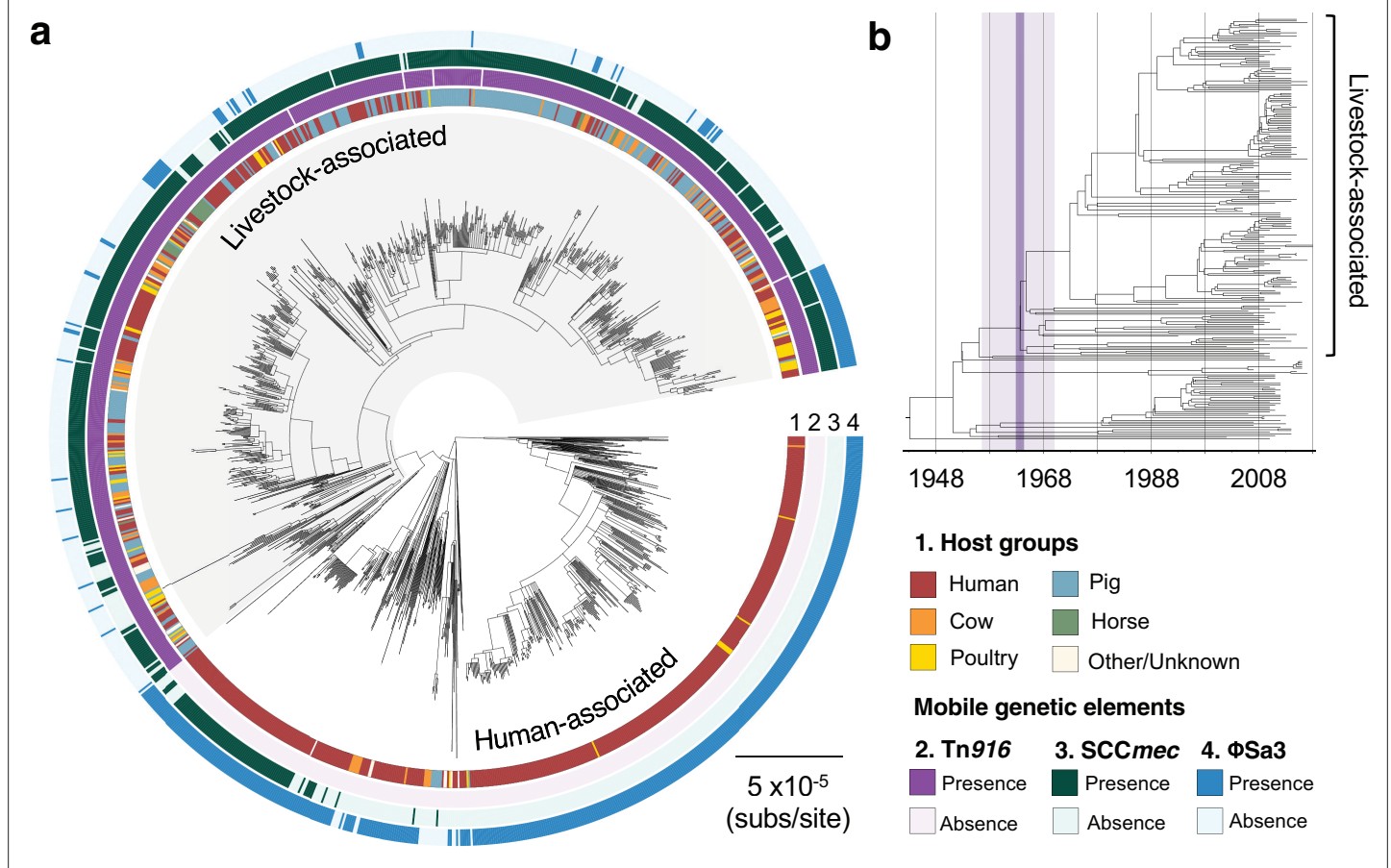

**Figure 1.** The transition to livestock association in the 1960s was accompanied by changes in the frequencies of three mobile genetic elements (MGEs). (**a**) A maximum likelihood phylogeny of 1180 isolates of CC398, rooted using an outgroup from ST291. Grey shading indicates the livestock-associated clade. Outer rings describe (1) the host groups isolates were sampled from, and the presence of three MGEs: (2) a Tn*916* transposon carrying *tetM*, (3) a SCC*mec* carrying *mecA*, and (4) a φSa3 prophage carrying a human immune evasion gene cluster. (**b**) A dated phylogeny of a sample of 250 CC398 isolates that shows livestock-associated CC398 originated around 1964 (95% HPD: 1957–1970).

The online version of this article includes the following figure supplement(s) for figure 1:

**Figure supplement 1.** The temporal, host species, and geographic distribution of our collection of CC398 isolates.

**Figure supplement 2.** Different outgroups consistently identify the root of CC398 within human-associated CC398.

**Figure supplement 3.** A consistent estimate of the age of the livestock-associated clade.

**Figure supplement 4.** Evidence of temporal signal is present across in our subsampled datasets, but is stronger when isolates from the human-associated group are excluded.

**Figure supplement 5.** Livestock- and human-associated CC398 have divergent accessory genomes, and genes whose presence/absence most clearly distinguish these groups (except one) are associated with a Tn*916* transposon, SCC*mec*, and φSa3 prophages.

*2009*; *Roberts and Mullany, 2009*). Several lines of evidence indicate that the presence of the Tn*916* transposon in livestock-associated CC398 is the result of a single acquisition event, followed by stable inheritance. First, the location of the transposon in the genome of livestock-associated CC398 is conserved (Tn*916* is always found next to the same core gene; WP_000902814 in the published annotation of S0385). Second, an alignment of the coding regions of this element (extracted from *de novo* assemblies) shows a similar average nucleotide diversity to core genes (*Figure 2d*). Third, a phylogeny constructed from the genes in this element is entirely congruent with the phylogeny of livestock-associated CC398 (*Figure 2b, c*).

Our analyses indicate that the Tn*916* transposon has been maintained in livestock-associated CC398 since its origin, and therefore for around 57 years (*Figure 1b*). Nevertheless, its absence from 5/704 livestock-associated CC398 isolates in our collection suggests that it remains capable of excision (this

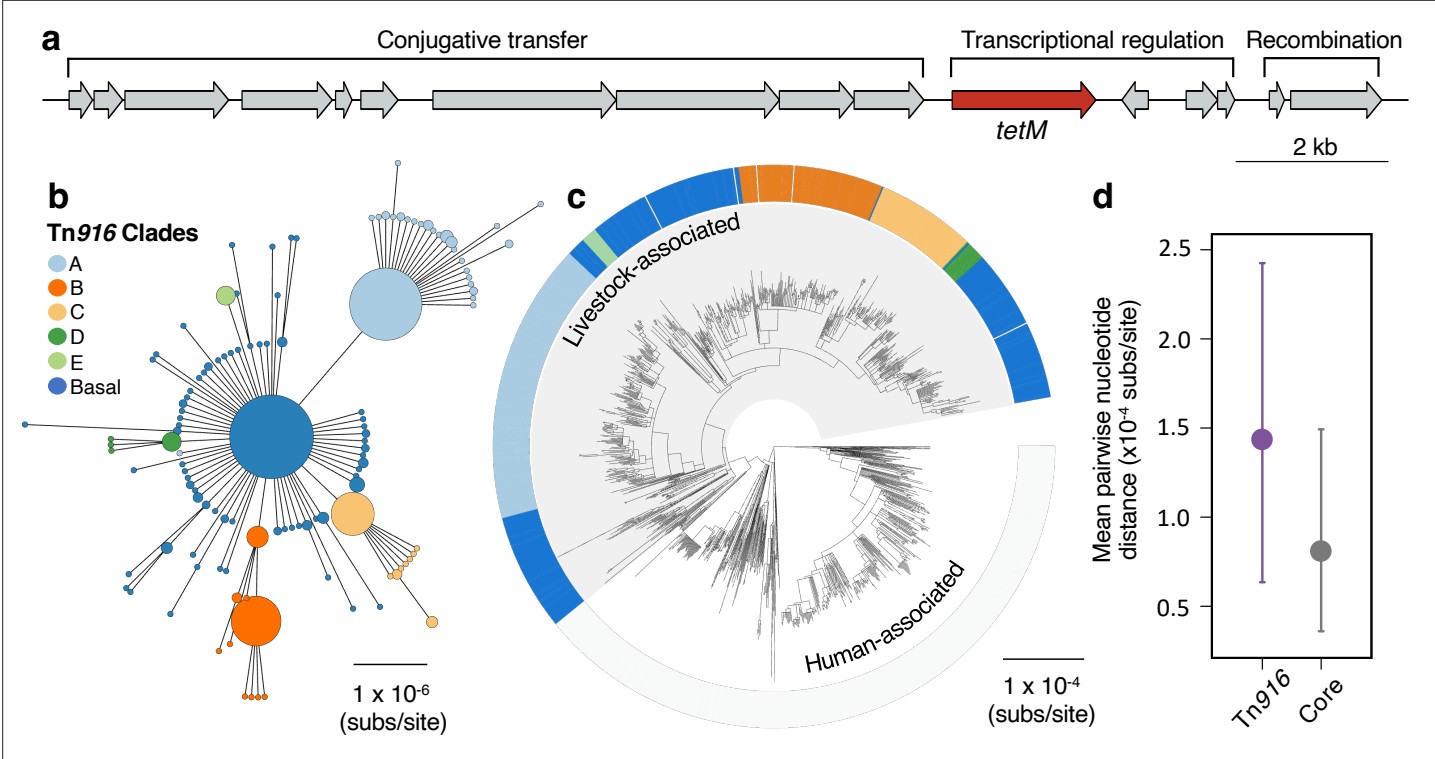

**Figure 2.** A Tn*916* transposon carrying *tetM* has been stably maintained by livestock-associated CC398 since its origin. (**a**) A gene map of the Tn*916* transposon in CC398 (based on reference genome 1_1439), with annotations based on previous studies (*de Vries et al., 2009*; *Roberts and Mullany, 2009*). (**b**) A minimum-spanning tree of the element based on a concatenated alignment of all genes shown in (**a**). Points represent groups of identical elements, with point size correlated with number of elements on a log scale, and colours representing well-supported clades (>70 bootstrap support in a maximum likelihood phylogeny) that include >10 elements (smaller clades are incorporated into their basal clade). (**c**) These clades are annotated onto the CC398 phylogeny as an external ring. (**d**) Mean pairwise nucleotide distance between isolates carrying the Tn*916* transposon based on genes in the Tn*916* transposon and core genes, using bootstrapping to estimate error (see Materials and methods for details).

The online version of this article includes the following figure supplement(s) for figure 2:

**Figure supplement 1.** Evidence of repeated excision of Tn*916* transposon in the livestock-associated clade.

is consistent with experimental studies that have shown that Tn*916* in CC398 is a functional conjugative transposon; *de Vries et al., 2009*). None of the genes associated with this element are present in these five isolates, and in two we were able to identify an intact integration site in the assembled genome (*Figure 2—figure supplement 1*). These five isolates are broadly distributed across the livestock-associated clade, and not linked to a particular host species or geographic location.

## More variable maintenance of a SCC*mec* carrying *mecA*, *tetK*, and *czrC*

Previous studies suggest that LA-MRSA CC398 emerged from human-associated methicillin-sensitive *S. aureus* (MSSA) (*Price et al., 2012*). However, the presence of SCC*mec* elements in recently sampled human-associated CC398 isolates that fall basal to the livestock-associated clade, including clinical isolates from China (*He et al., 2018*; *Zou et al., 2022*), Denmark (*Moller et al., 2019*), and New Zealand (*Gonçalves da Silva et al., 2017*), makes the association between methicillin resistance and livestock association less clear (*Figure 1a*).

SCC*mec* elements in *S. aureus* are categorised into several types (*Hanssen and Ericson Sollid, 2006*). Consistent with previous studies of CC398 (e.g. *Price et al., 2012*), we observe both type V (76%) and type IV (21%) in CC398 (we were unable to confidently type the remaining 3%; *Figure 3c*, *Figure 3—figure supplement 1*, and *Supplementary files 3 and 5*). Most of the type V SCC*mec* elements belong to the subtype Vc previously described in livestock-associated CC398 (*Li et al., 2011*; *Price et al., 2012*; *Vandendriessche et al., 2014*). This element includes two additional resistance genes: *tetK* (tetracycline resistance) and *czrC* (heavy metal resistance) (*Figure 3a* and *Supplementary file 6*). We identified a full-length version of this element in 335 genomes (including some

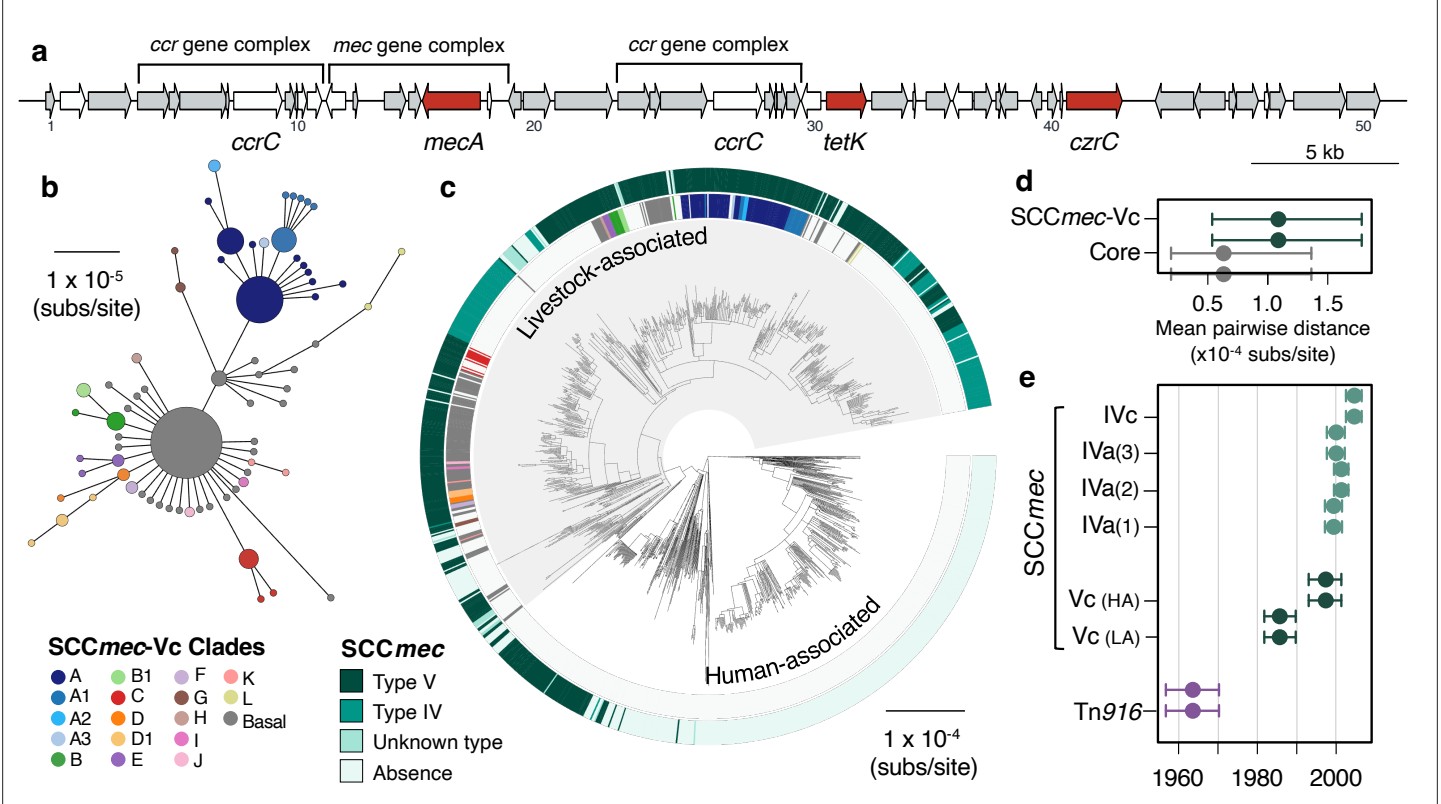

**Figure 3.** A type V SCC*mec* has been maintained since the 1980s, with occasional replacements. (**a**) A gene map of the type Vc *SCCmec* element in CC398 (using the 1_1439 reference strain), with annotations from previous studies (*Li et al., 2011*; *Vandendriessche et al., 2014*). Genes in white were excluded from analyses of diversity within the element due to difficulties in distinguishing homologues. (**b**) A minimum-spanning tree of the type Vc *SCCmec* element based on a concatenated alignment of the genes (grey and red) in (**a**). Points represent groups of identical elements, point size correlates with group size on a log scale, and colours represent well-supported clades (>70 bootstrap support in a maximum likelihood phylogeny). (**c**) Well-supported clades and *SCCmec* type are annotated on the CC398 phylogeny in external rings. (**d**) Mean pairwise nucleotide distance between isolates carrying the *SCCmec* type Vc based on genes in the *SCCmec* type Vc and core genes, with error estimated by bootstrapping (see Materials and methods for details). (**e**) Acquisition dates for different *SCCmec* elements and Tn*916* inferred from an ancestral state reconstruction over the dated phylogeny in 1 (**b**). Dates for type Vc are shown for both livestock- and human-associated CC398.

The online version of this article includes the following figure supplement(s) for figure 3:

**Figure supplement 1.** Type V and IV *SCCmec* elements identified in CC398 through a *BLASTn* search or representative types.

**Figure supplement 2.** Most of the shorter versions of the type Vc *SCCmec* element in CC398 can be attributed to deletion events.

**Figure supplement 3.** There have been at least four independent acquisitions of type IV *SCCmec* within livestock-associated CC398.

**Figure supplement 4.** Tn*916* was acquired before the current complement of *SCCmec* elements in livestock-associated CC398, *SCCmec* type V was acquired before *SCCmec* type IV, and *SCCmec* type V was acquired by livestock-associated CC398 before human-associated CC398.

of the most recent isolates in our collection, sampled from UK pig farms in 2018) and shorter type V elements in 204 genomes. Full-length versions are only observed in the livestock-associated clade, while shorter versions are found in both livestock- and human-associated groups of CC398. Shorter versions often lacked *tetK* (*n* = 90) and *czrC* (*n* = 117) genes (*Figure 3—figure supplement 2*). Type IV elements are only observed in the livestock-associated clade. They include subtypes IVa and IVc, and all only carry a single resistance gene: *mecA*.

While type Vc is the most common *SCCmec* in our collection of livestock-associated CC398, this largely reflects isolates from pigs. Type Vc *SCCmec* is much more common in pigs (77% of isolates; *n* = 286) than type IV *SCCmec* (7% of isolates). In cows (*n* = 74), the difference is reduced: 45% are type Vc and 32% are type IV. And in isolates from other animal species (*n* = 94) type IV elements (55%) are more common than type Vc (33%).

Diversity within the type Vc *SCCmec* element indicates that a full-length type Vc *SCCmec* was acquired once by livestock-associated CC398, and has been maintained within CC398 largely through

vertical transmission. First, low nucleotide diversity within full-length versions of the element is consistent with 329/335 sharing a recent origin common within livestock-associated CC398 (*Figure 3d*). Second, patterns of diversity are largely congruent with the core genome phylogeny, consistent with vertical inheritance (*Figure 3b, c*). Third, low nucleotide diversity within genes shared across full- and shorter-length versions of the element are consistent with most shorter-length versions being the result of deletion within livestock-associated CC398 (*Figure 3—figure supplement 2*). In contrast, diversity in the type IV elements in livestock-associated CC398 supports four independent acquisitions from outside of CC398 (*Figure 3—figure supplement 3*). Similar to the type Vc element, once acquired, these elements tend to be maintained.

While the SCC*mec* elements carried by human-associated CC398 are always type V, 70/80 fall within a single clade from a hospital outbreak in Denmark in 2016 (*Moller et al., 2019*). They show a truncation relative to the full-length type Vc that is also observed in three isolates from livestock-associated CC398 (leading to the absence of *czrC*, *Figure 3—figure supplement 2*). Pairwise nucleotide distances between these 70 human-associated CC398 elements and the full-length type Vc in livestock-associated CC398 are consistent with a recent common ancestor within CC398. In contrast, nucleotide diversity within the other 10 type V SCC*mec* in human-associated CC398 and distances from the livestock-associated CC398 type Vc indicate multiple independent acquisitions from outside of CC398 (*Figure 3—figure supplement 2*).

Using our dated phylogeny of CC398 and categorisation of SCC*mec* based on diversity within the elements, we inferred the dynamics of gain and loss within CC398 (*Figure 3e* and *Figure 3—figure supplement 4*). These reconstructions consistently estimated that the type Vc SCC*mec* had been acquired by livestock-associated CC398 by around 1986 (95% CI: 1982–1990), and has therefore been maintained within livestock-associated CC398 for around 35 years. While the diversity within this element indicates a single acquisition by CC398, these reconstructions indicate multiple gains, likely reflecting horizontal transmission within CC398. They also indicate that the acquisition by livestock-associated CC398 predated the acquisition by human-associated CC398, consistent with transmission from livestock-associated CC398 to human-associated CC398. In contrast, type IV elements show evidence of several more recent acquisitions in relatively quick succession between 1997 and 2004 (*Figure 3e* and *Figure 3—figure supplement 4*).

Together, our analyses are consistent with LA-MRSA emerging from human-associated MSSA and acquiring the type Vc SCC*mec* element following its initial diversification. The element has been stably maintained (particularly in pigs), although with several exceptions – including deletions of parts of the element and replacements with smaller type IV SCC*mec*.

## Loss of a φSa3 prophage carrying a human immune evasion gene cluster, but with frequent reacquisition

In contrast to the maintenance of Tn*916* and SCC*mec* type V in the livestock-associated clade, the φSa3 prophage is highly dynamic. φSa3 prophages carry human immune evasion gene clusters that include a variable set of functional genes that encode human-specific virulence factors, including *sak*, *chp*, *scn*, *sea*, and *sep* (*Gladysheva et al., 2003*; *Postma et al., 2004*; *Rooijakkers et al., 2005*; *van Wamel et al., 2006*; *Thammavongsa et al., 2015*; *van Alen et al., 2018*). They are temperate prophages that primarily integrate into the *hlb* gene of *S. aureus*. While φSa3 prophages are present in 88% of the human-associated CC398 isolates in our collection, we find that this is not a consequence of a stable association between CC398 and one prophage. Nucleotide diversity within genes shared across φSa3 prophages (*Figure 4* and *Supplementary file 3*) suggest at least seven (but likely more) acquisitions of an φSa3 prophage into human-associated CC398 from outside of CC398. The set of functional genes carried by these elements indicate that the elements within human-associated CC398 include types C (*n* = 285), B (*n* = 111), E (*n* = 35), A (*n* = 4), and D (*n* = 4), and those carried by livestock-associated CC398 isolates include types B (*n* = 84), E (*n* = 8), and A (*n* = 5) (*van Wamel et al., 2006*; *van Alen et al., 2018*).

While φSa3 prophages are rare in livestock-associated CC398 (68/704 isolates), diversity within these elements indicate at least 15 (but likely more) acquisitions of these MGEs by livestock-associated CC398. The majority of these elements (69%) do not share a recent common ancestor with those in human-associated CC398. φSa3 prophages present in livestock-associated CC398 generally show evidence of recent acquisition, with a notable exception being a previously described

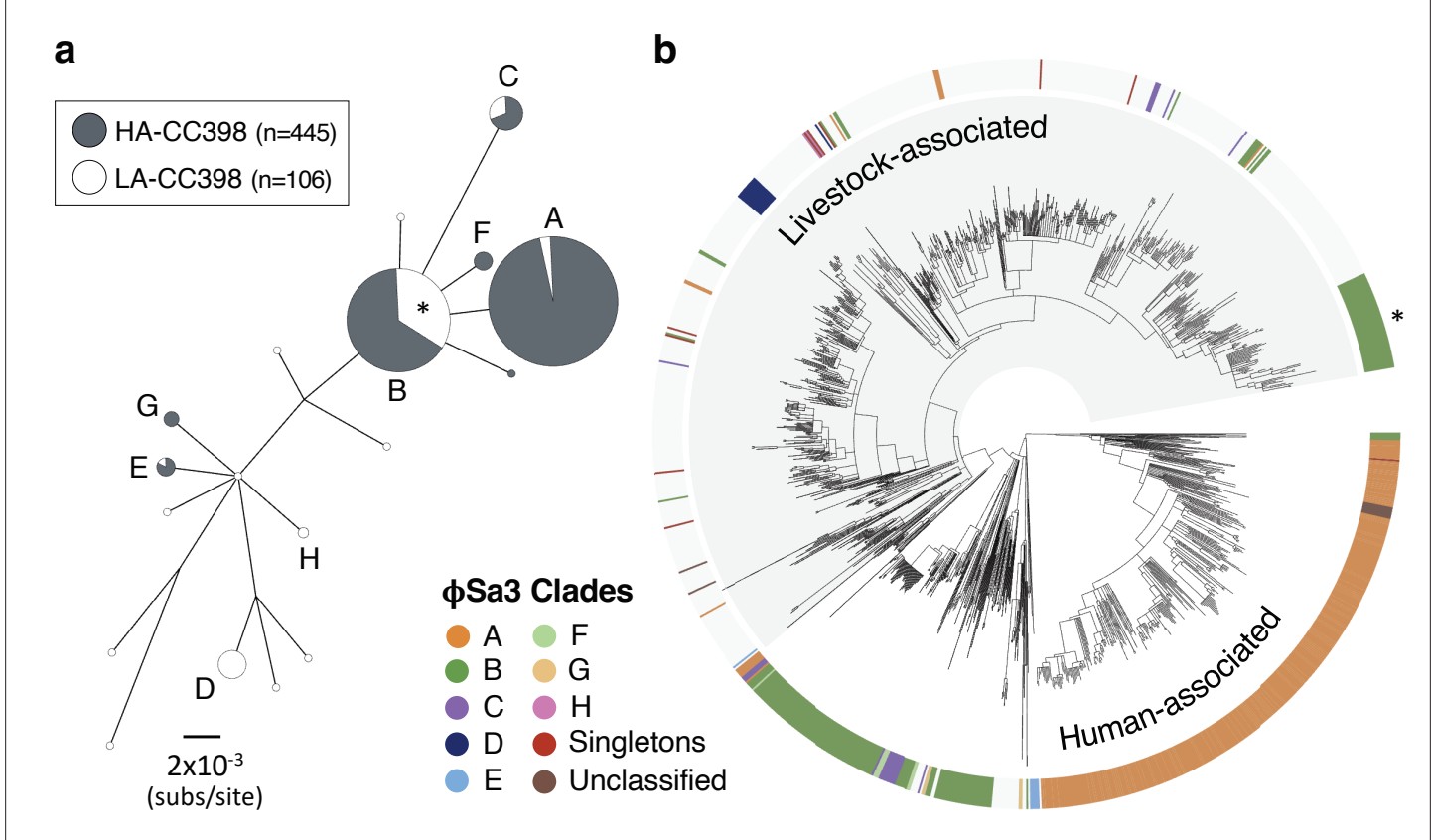

**Figure 4.** φSa3 prophages have been lost and acquired multiple times in both human- and livestock-associated CC398. (**a**) A maximum likelihood phylogeny based on 12 genes shared across the φSa3 prophages in our collection, with both low-support nodes (<70% bootstrap support) and branches <0.0018 subs/site collapsed. The latter cut-off is a conservative estimate of the maximum distance that could reflect divergence within CC398. It is the maximum pairwise distance between isolates carrying φSa3 prophages across 1000 estimates from random samples of a core gene alignment of the same number of sites as is in our φSa3 prophage alignment. Node size correlates with the number of elements on a log scale. Elements carried by isolates from human-associated CC398 (grey) and livestock-associated CC398 (white) isolates are indicated, and nodes that include multiple elements labelled (**A–E**). (**b**) These clades annotated on the CC398 phylogeny as an external ring. The element carried by the poultry-associated subclade of livestock-associated CC398 is indicated by *.

The online version of this article includes the following figure supplement(s) for figure 4:

**Figure supplement 1.** Maintenance of a φSa3 prophage in a poultry-associated clade of livestock-associated CC398 for around 21 years.

poultry-associated subclade (*n* = 51) (*Price et al., 2012*; *Larsen et al., 2016b*; *Pérez-Moreno et al., 2017*; *Tang et al., 2017*). In our collection, 39% of the isolates in this clade are from poultry (compared to 4% across the rest of the livestock-associated clade). Low nucleotide diversity within the type B φSa3 consistently present in isolates within this clade suggests that it has been maintained since its acquisition approximately 21 years ago (95% CI: 1997–2001; *Figure 1b* and *Figure 4—figure supplement 1*).

## Distinct route to multi-drug resistance in livestock-associated CC398

Livestock-associated CC398 is frequently multi-drug resistant. We see evidence of this in our dataset where 81% of livestock-associated CC398 isolates carry one or more resistance genes for antibiotic classes other than tetracyclines and β-lactams (*Figure 5—figure supplement 1*, *Figure 5—figure supplement 2*, and *Supplementary file 7*). Sixty-seven percent of livestock-associated CC398 isolates have genes associated with trimethoprim resistance (*dfrA*, *dfrK*, or *dfrG*), 42% have genes associated with macrolide resistance (*ermA*, *ermB*, *ermC*, or *ermT*), and 26% have genes associated with amino-glycoside resistance (*aadA*, *aphA*, or *aphD*). Not only are resistance genes less common in human-associated CC398 (only 20% of isolates carry tetracycline resistance genes, and 9% carry trimethoprim

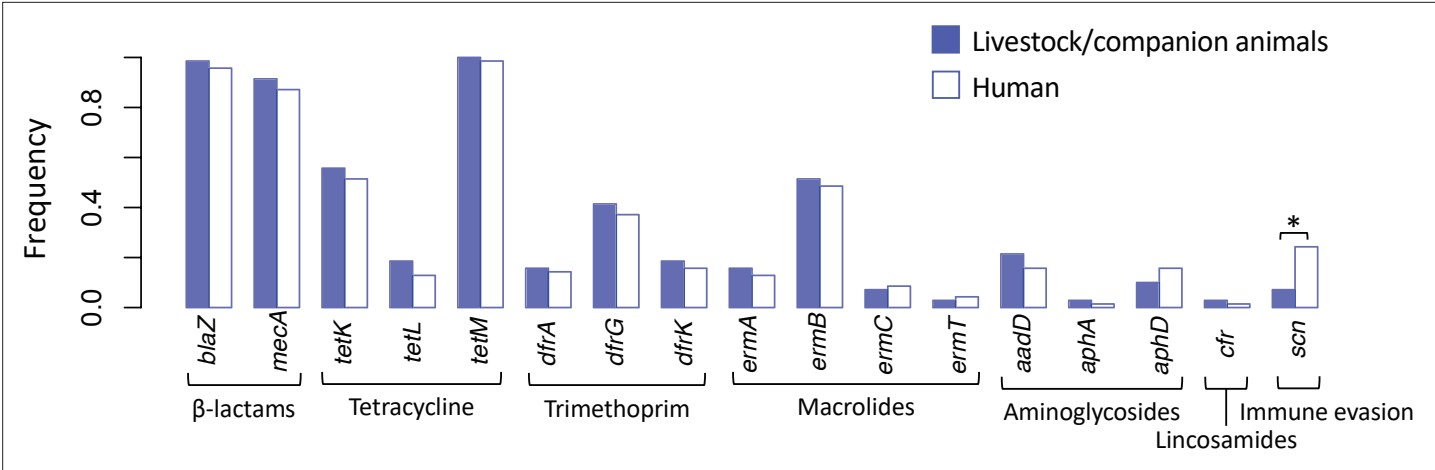

**Figure 5.** Spillover of livestock-associated CC398 into humans is associated with acquisition of human immune evasion genes. Seventy phylogenetically independent clades that include isolates from both humans and other species were identified within the livestock-associated clade. The plot shows the frequency with which these genes were identified within isolates from humans (right, empty bars) and non-human species (left, filled bars) in these groups. An asterisk indicates a significant difference based on McNemar's chi-squared test ($p = 1.50 \times 10^{-3}$). No resistance genes differed significantly in their frequency across the human and non-human hosts ($p > 0.1$). The *scn* gene is always present in the human immune evasion cluster carried by φSa3 prophages, and therefore represents the presence of this element.

The online version of this article includes the following figure supplement(s) for figure 5:

**Figure supplement 1.** Patterns in the presence/absence of antibiotic resistance genes vary across livestock- and human-associated CC398 in addition to the three genes associated with Tn*916* and SCC*mec*.

**Figure supplement 2.** Differences in antibiotic resistance gene frequencies across human- and livestock-associated CC398.

**Figure supplement 3.** Phylogeny of the livestock-associated clade showing locations of 70 phylogenetically independent groups of isolates from human (red) and livestock/companion animal (blue) hosts.

resistance genes), they also differ in their relative frequencies. In particular, human-associated CC398 isolates more commonly carry genes associated with macrolide resistance (91%).

## Spillover of livestock-associated CC398 into humans is associated with the acquisition of φSa3 prophages, but not a loss of resistance genes

φSa3 prophages are more common in human isolates (23%) than in livestock or companion animal isolates (11%) in our collection of livestock-associated CC398. To determine the significance of this association, we identified 70 phylogenetically independent clades that include isolates from both human and livestock or companion animal hosts (*Figure 5—figure supplement 3*). Comparisons across these groups revealed that isolates from humans are consistently more likely to carry an φSa3 prophage (McNemar's chi-squared test: $p = 1.50 \times 10^{-3}$; *Figure 5*).

φSa3 prophages are also less common in isolates from pigs than from other non-human species (only 3% of pig isolates carry one). In 42/70 of our phylogenetically independent groups, the only non-human species was a pig, and none of the pig isolates from these groups carried an φSa3 prophage (while 14% of the human isolates did). In the remaining 28 groups (that included isolates from cows, horses, and poultry), φSa3 prophages were observed in non-human hosts in 17% of groups, but still at a higher frequency in humans (39% of groups) (McNemar's chi-squared test: $p = 0.04$).

In contrast, we found no evidence that the spillover of livestock-associated CC398 into humans is associated with the loss (or gain) of individual antibiotic resistance genes (*Figure 5*). This contrasts with the conclusions of a previous study that suggested that the resistance genes carried by livestock-associated CC398 are likely to be lost in human hosts (*Sieber et al., 2019*). We find that no resistance gene was significantly more or less common in humans than in livestock species (McNemar's chi-squared tests: $p > 0.1$), nor was there a consistent shift in the overall number of resistance genes (there were more resistance genes in isolates from human hosts in 22/70 pairs and more in livestock hosts in 34/70 pairs).

## Discussion

We have characterised the evolutionary dynamics of the three classes of MGEs that show the greatest changes in frequency across human- and livestock-associated CC398: the Tn*916* transposon and the SCC*mec* element, which are both common in livestock-associated CC398, and the φSa3 prophage, which is common in human-associated CC398. Despite a consistency in the relative frequencies of these elements across CC398, leading to their strong association with the transition to livestock, these three elements show a broad spectrum of dynamics. The Tn*916* transposon carrying *tetM* shows evidence of stable and consistent vertical transmission in livestock-associated CC398 and absence from human-associated CC398. The type Vc SCC*mec*, carrying not only *mecA*, but also *tetK* and *czrC*, has also been stably maintained by several lineages of livestock-associated CC398, but by a combination of vertical and horizontal transmission, and with occasional replacement with type IV elements. While type V SCC*mec* elements are also present in human-associated CC398, there is little evidence of their longer-term maintenance. Finally, while φSa3 prophages carrying a human immune evasion gene cluster are rare in livestock-associated CC398 and common in human-associated CC398, there have been frequent gains and losses in both groups. These contrasting dynamics may reflect variation in the selective benefits and costs of the carriage of these MGEs by CC398, their availability in the environments encountered by CC398, and their mechanisms of transfer.

The three classes of MGEs whose dynamics we describe all employ different mechanisms of horizontal transfer, and this may both influence their intrinsic stability in the CC398 genome and their availability for acquisition from outside of CC398. Experimental studies have found that different types of MGEs vary in their rates of transfer between bacterial cells. In particular, *in vitro* rates of transfer of phage have been found to be several orders of magnitude higher than of transposons (*Humphrey et al., 2021*). A 16-day *in vivo* study of MGE dynamics for two strains of CC398 also observed variation in the mobility of MGEs: the Tn*916* transposon, the type V SCC*mec* and an φSa3 prophage were all stably maintained and not horizontally transferred, while other prophages and plasmids were both gained and lost by daughter cells (*McCarthy et al., 2014*). While variation in the intrinsic stability of MGEs in their host will influence their long-term dynamics, other factors may dominate. For instance, the long-term stability of Tn*916* within CC398 could emerge from a broad spectrum of short-term dynamics. It could result from a very low rate of loss, or frequent loss combined with a high selective cost of loss and a low probability of reacquisition from sources other than very closely related cells. Therefore, the long-term dynamics we describe provide a unique insight into the relationship between these MGEs and CC398.

We find that a Tn*916* transposon that carries *tetM* was acquired once by CC398 and this coincided with its origin in European livestock in 1964 (95% CI: 1957–1970). Our observation of the loss of Tn*916* on terminal branches of the phylogeny suggests that while losses do occur, lineages that lose Tn*916* are either rapidly outcompeted by those that have maintained it, or they rapidly reacquire it from close relatives. Antibiotics, including tetracyclines, were first licensed for use as growth promoters in livestock in European countries in the 1950s, and were in common use by the end of that decade (*Lowbury, 1958*). While the use of antibiotics as growth promoters was banned by the European Union in 2006, tetracyclines remain the most commonly used antimicrobial class in livestock farming (*World Health Organization, 2015*; *DANMAP, 2019*). Carriage of Tn*916* is therefore likely to have been associated with a strong selective benefit for livestock-associated CC398 ever since its emergence. However, as CC398's exposure to tetracyclines will be intermittent, the long-term stability of Tn*916* is also likely to reflect a low selective cost in the absence of treatment or a barrier to reacquisition following loss. Tn*916*-like elements are found across several bacterial genera (*Clewell et al., 1995*; *Roberts and Mullany, 2009*), including other opportunistic pathogens in the respiratory microbiome of pigs (*Holden et al., 2009*; *Hoa et al., 2011*) and other *S. aureus* CCs (*de Vries et al., 2009*). This suggests that the stability of Tn*916* in CC398 is not due to the rarity of the element in the environments encountered by CC398, however it may be due to other barriers to successful transfer.

Previous studies have found evidence that the regulatory system of Tn*916* promotes its maintenance in a host through ensuring both that excision only occurs in the presence of tetracycline (or other transcription-limiting cell stress) and that any selective burden in the absence of treatment is minimised (*Roberts and Mullany, 2009*). Nevertheless, Tn*916* has been found to have a much more dynamic association with lineages of other bacterial species (e.g. *Streptococcus pneumoniae* [*D'Aeth et al., 2021*]). The fitness costs of MGEs can be host (and even insertion locus) specific and can also be

mitigated over time (*Starikova et al., 2013*; *Durão et al., 2018*), and therefore the stability of Tn*916* in livestock-associated CC398 might reflect a low cost that is specific to this lineage. A combination of high selective benefit, low selective cost and inaccessibility for reacquisition following loss could explain the remarkable stability of Tn*916* in livestock-associated CC398, and may in part explain the success of this lineage in livestock.

Our results indicate that the acquisition of an SCC*mec* occurred later in the expansion of livestock-associated CC398. The most common SCC*mec* in livestock-associated CC398, type Vc, carries *tetK* and *czrC* in addition to *mecA*. These additional resistance genes might be highly advantageous in livestock, and particularly in pigs. While all livestock-associated CC398 carry *tetM*, there is evidence that carrying *tetK* in addition to *tetM* is associated with increased fitness during exposure to sublethal concentrations of tetracycline (*Larsen et al., 2016a*). *czrC* is associated with heavy metal resistance (*Cavaco et al., 2010*), which is likely to be beneficial in the context of the common supplementation of animal feed with zinc oxide, which in pigs is commonly used to prevent diarrhoea in weaners (*Nielsen et al., 2021*). Additionally, *mecA* is likely to be beneficial because of the common use of β-lactams in livestock farming, including third generation cephalosporins (*Sjölund et al., 2016*; *Lekagul et al., 2019*). The size of the type Vc element, combined with the replacements and truncations we observe suggests that it may come with a selective cost. In addition, SCC*mec* type Vc appears to be rare (at least in *S. aureus*). The element has been found in other staphylococci (*S. cohhii* in Vervet Monkeys [*Hoefer et al., 2021*]), but has not been reported in other *S. aureus* CCs.

Loss of the type Vc SCC*mec* on internal branches within livestock-associated CC398 is associated with replacement with a type IV SCC*mec* that only carries *mecA*. While we find evidence of only a single acquisition of the type Vc SCC*mec* by livestock-associated CC398, we find evidence of at least four acquisitions of type IV SCC*mec*. The more recent dates of acquisition, and an apparent association with livestock species other than pigs, might reflect a difference in selective pressures across different livestock species, or a loss of the type Vc element during transmission between livestock populations. The long-term maintenance of the type Vc SCC*mec* is consistent with a high selective benefit (particularly in pigs where this element is most common). However, its repeated loss and replacement with type IV elements are consistent with a low cost of this replacement, at least in certain contexts (perhaps in other livestock hosts), and might also reflect the rarity of the type V element relative to the type IV.

While the loss of the φSa3 prophage that carries a human immune-evasion gene cluster is associated with the transition to livestock, neither the loss nor the gain of these elements is likely to be a substantial hurdle for the adaptation of CC398 to human or non-human hosts. Their ubiquity in human-associated CC398 and frequent acquisition following transmission of livestock-associated CC398 to humans (consistent with *Sieber et al., 2019*), is consistent with a strong benefit in the human host (*Rohmer and Wolz, 2021*). On the other hand, as we find that these elements are frequently lost and generally absent in other host species, these elements may carry a selective burden outside of the human host. The diversity of φSa3 prophages in CC398 suggests they form a large pool of elements within human hosts (*van Alen et al., 2018*), likely reflecting their ubiquity in human carriage populations of *S. aureus* (*Rohmer and Wolz, 2021*) and capacity to transfer between *S. aureus* lineages.

We observe one clear exception to the pattern of recent acquisitions of φSa3 in livestock-associated CC398 in response to spillover events: the acquisition of an φSa3 prophage at the base of a poultry-associated subclade of livestock-associated CC398 (*Price et al., 2012*; *Larsen et al., 2016b*; *Pérez-Moreno et al., 2017*; *Tang et al., 2017*). This clade was first described as a hybrid LA-MRSA CC9/CC398 lineage (*Price et al., 2012*), and was subsequently investigated as a lineage associated with human disease (*Larsen et al., 2016b*). The maintenance of an φSa3 in this lineage may reflect more frequent transmission via a human host, or an adaptation to poultry (a recent study suggested that φSa3 prophages might aid immune evasion in species other than humans [*Jung et al., 2017*]). Either way, this might make this lineage a greater immediate threat to public health.

While livestock-associated CC398 is found across a broad range of livestock species, it is most commonly associated with pigs. The dynamics of the SCC*mec* and φSa3 prophages that we have identified are both consistent with livestock-associated CC398 originating in pig farms and later spreading to other livestock species. The lower frequency of the type Vc SCC*mec* in species other than pigs could reflect random loss during transmission bottlenecks, or a reduced benefit of either *tetK* or *czrC*

in these species. Similarly, the higher frequency of φSa3 prophages in other species might reflect an increased benefit or reduced cost, or more recent transmission via human hosts.

Our results reveal that LA-MRSA CC398 is a stably antibiotic-resistant pathogen that is capable of dynamic readaptation to humans. We find that Tn*916* and SCC*mec* are both stably maintained in livestock-associated CC398, across different livestock species and countries, and that neither of these MGEs (or other antibiotic resistance genes) tend to be lost when livestock-associated CC398 is transmitted to humans. This suggests that these MGEs are associated with a low selective cost in both livestock and human hosts, and therefore across variable levels and types of antibiotic exposure. The stability of these two MGEs, combined with the capacity of livestock-associated CC398 to rapidly acquire the φSa3 prophage, underlines the threat posed by LA-MRSA to public health. While SCC*mec* is less stably maintained than Tn*916*, our identification of several independent acquisitions of type IV SCC*mec* suggests both a strong selective benefit, not contingent on the carriage of *tetK* and *czrC*, and an availability of type IV elements. These dynamics predict that the impact of gradual reductions in antibiotic consumption on LA-MRSA is likely to be slow to be realised and that the forthcoming EU ban on medical zinc supplementation in pig feed may have a limited impact on LA-MRSA (*European Medicines Agency, 2017*; *European Medicines Agency, 2020 DANMAP, 2019*). Further work is, however, required to understand the factors that underlie the acquisition and maintenance of resistance genes within LA-MRSA, and how they differ from human-associated MRSA lineages.

## Materials and methods

### Data collection

All available genome sequence data relating to *S. aureus* CC398 were downloaded from public databases (https://www.ncbi.nlm.nih.gov/sra and https://www.ebi.ac.uk/ena/browser/; accessed 2019), with metadata in some cases obtained by request (*Supplementary file 1*). We additionally sequenced five isolates sampled from UK pig farms in 2018. All publicly available complete *S. aureus* genomes assemblies (https://www.ncbi.nlm.nih.gov/; accessed 2019) were MLST typed using Pathogenwatch (https://pathogen.watch), and the 43 genomes identified as CC398 were added to our collection. After exclusion of low-quality assemblies and isolates mis-characterised as CC398, this led to a collection of 1,180 genomes.

### Genomic library preparation and sequencing

For sequencing of the five UK pig farm isolates, genomic DNA was extracted from overnight cultures grown in TSB at 37°C with 200 rpm shaking using the MasterPure Gram Positive DNA Purification Kit (Cambio, UK). Illumina library preparation and Hi-Seq sequencing were carried out as previously described (*Harrison et al., 2013*).

### Genome assembly

We used sequence data from all isolates to generate *de novo* assemblies with *Spades* v.3.12.0 (*Bankevich et al., 2012*). We removed adapters and low-quality reads with *Cutadapt* v1.16 (*Martin, 2011*) and *Sickle* v1.33 (*Joshi and Fass, 2011*), and screened for contamination using *FastQ Screen* (*Wingett and Andrews, 2018*). Optimal k-mers were identified based on average read lengths for each genome. All assemblies were evaluated using *QUAST* v.5.0.1 (*Gurevich et al., 2013*) and we mapped reads back to *de novo* assemblies to investigate polymorphism (indicative of mixed cultures) using *Bowtie2* v1.2.2 (*Langmead and Salzberg, 2012*). Low-quality genome assemblies were excluded from further analysis (i.e. N50 < 10,000, contigs smaller than 1 kb contributing to >15% of the total assembly length, total assembly length outside of the median sequence length ± one standard deviation, or >1500 polymorphic sites). We identified genomes mischaracterised as CC398 via two approaches and excluded them from further analysis. First, we identified STs with *MLST-check* (*Page et al., 2016*) and grouped into CCs using the *eBURST* algorithm with a single locus variant (*Francisco et al., 2009*). Second, we constructed a neighbour-joining tree based on a concatenated alignment of MLST genes (*arcC, aroE, glpF, gmk, pta, tpi,* and *yqiL*) for our collection and 13 additional reference genomes from other CCs, using the *ape* package in *R* and a K80 substitution model (*Paradis et al., 2004*).

We generated reference-mapped assemblies with *Bowtie2* using the reference genome S0385 (*GenBank* accession no. AM990992). For reference genomes, we generated artificial FASTQ files with *ArtificialFastqGenerator* (*Frampton and Houlston, 2012*). Average coverage and number of missing sites in these assemblies were used as an additional quality control measure; genomes with average coverage <×50 or with >10% missing sites were excluded.

We identified recombination in the reference-mapped alignment using both *Gubbins* v2.3.1 (*Croucher et al., 2015*) and *ClonalFrame* (*Didelot and Wilson, 2015*). We masked all the recombinant sites identified from our alignment. We additionally masked a region of ~123 kb that was identified as horizontally acquired from an ST9 donor in a previous study (*Price et al., 2012*).

## Genome annotation and identification of homologous genes

We annotated *de novo* assemblies with *Prokka* v2.8.2 (*Seemann, 2014*) and identified orthologous genes with *Roary* (*Page et al., 2015*) using recommended parameter values. We created a core gene alignment with *Roary* and identified recombinant sites using *Gubbins*. We identified antibiotic resistance genes using the *Pathogenwatch* AMR prediction module (Wellcome Sanger Institute), which uses *BLASTn* (*Altschul et al., 1990*) with a cut-off of 75% coverage and 80–90% identity threshold (depending on the gene) against a *S. aureus* AMR database.

## Phylogenetic analyses

We carried out phylogenetic reconstruction for the reference-mapped alignment with *RAxML* v8.2.4 using the $GTR+\Gamma$ model and 1000 bootstraps (*Stamatakis, 2014*). Sites where >0.1% of genomes showed evidence of recombination or had missing data were excluded from the analysis. We constructed dated phylogenies using *BEAST* v1.10 with a $HKY+\Gamma$ model, a strict molecular clock, and constant population size coalescent prior, from the coding regions of the reference-mapped alignment (*Drummond et al., 2012*). We fit separate substitution models and molecular clocks to first/second and third codon positions to reflect differences in selective constraint. We constructed phylogenies for three random subsamples of 250 isolates (200 from livestock-associated CC398 and 50 from human-associated CC398). Subsamples were non-overlapping except for 30 genomes representing the most divergent lineages within the livestock-associated clade, to ensure a consistent description of the origin of this clade. We constructed additional phylogenies from subsamples that included only isolates from the LA clade to establish that consistent rate estimates were returned over different evolutionary depths. We investigated temporal signal in each dataset through a regression of root-to-tip distance against sampling date, and a permutation test of dates over tips (with clustering used to correct for any confounding of temporal and genetic structures) (*Murray et al., 2016*; *Rambaut et al., 2016*). Trees are visualised and annotated using *ITOL* (*Letunic and Bork, 2021*).

## Comparative analyses of MGEs

Genes associated with the transition to livestock were identified through comparing frequencies of homologous genes identified with *Roary* across human- and livestock-associated CC398. We investigated the association between these genes and MGEs through analysis of (1) physical locations within our *de novo* assemblies and the reference genomes, (2) correlations in their presence/absence across our collection, and (3) comparison with descriptions in the literature and in online databases of particular MGEs.

We confirmed the identity of the Tn*916* transposon by comparison with published descriptions and publicly available annotated sequences. SCC*mec* elements were initially categorised into types using a *BLASTn* search of all representative SCC*mec* types from the *SCCmecFinder* database in our *de novo* assemblies (*Supplementary file 5*; *Hanssen and Ericson Sollid, 2006*). φSa3 prophages were initially identified and categorised through identification of functional genes associated with the human immune evasion gene cluster. To further characterise diversity within these elements we identified genes associated with each element across our collection. We constructed alignments of genes within MGEs using *Clustal Omega* v1.2.3 (*Sievers and Higgins, 2018*) and checked for misalignment by eye.

We analysed variation in both gene content and nucleotide diversity within shared genes for each MGE. We estimated pairwise nucleotide distances in concatenated alignments of shared genes for each type of element using the *ape* package in *R* (*Paradis et al., 2004*), and constructed maximum

likelihood trees using *RaxML* and minimum-spanning trees using *GrapeTree* (*Zhou et al., 2018*) to investigate co-phylogeny. We generated confidence intervals for our estimates of mean pairwise nucleotide distances within MGEs by re-estimating the mean distance from 1000 bootstrapped samples of sites. We compared this to sites in the core genome by sampling the same number of sites as were in the MGE alignment from a concatenated alignment of core genes (generated by *Roary*). For SCC*mec* we used ancestral state reconstruction in *BEAST* to infer the evolutionary dynamics of these elements and date their origins within CC398. This involved fitting a discrete traits model to the posterior distributions of trees, with each state representing a version of the element that had been independently acquired by CC398. We used a strict clock model that allowed for asymmetric rates of transitions between states, but we found that the results were robust to use of a symmetric model or a relaxed clock.

## Phylogenetically independent groups

To test the association between spillover into the human host and the presence of φSa3 prophages and antibiotic-resistance genes, we identified 70 phylogenetically independent clades of isolates that were sampled from both human and non-human hosts in the livestock-associated clade. We classed a gene as present in a host if it was observed in any of the isolates from that host in that clade.

## Acknowledgements

MM was funded by the Medical Research Council, co-funded by the Raymond and Beverly Sackler Fund. GGRM and LAW were supported by a Sir Henry Dale Fellowship jointly funded by the Wellcome Trust and the Royal Society (109385/Z/15/Z). GGRM was also supported by a ZELS BBSRC award (BB/L018934/1) and a Research Fellowship at Newnham College.

## Additional information

### Funding

| Funder | Grant reference number | Author |
| --- | --- | --- |
| Wellcome Trust | 109385/Z/15/Z | Gemma GR Murray<br>Lucy A Weinert |
| Medical Research Council | | Marta Matuszewska |
| Biotechnology and Biological Sciences Research Council | BB/L018934/1 | Gemma GR Murray |
| Newnham College, University of Cambridge | | Gemma GR Murray |

The funders had no role in study design, data collection, and interpretation, or the decision to submit the work for publication. For the purpose of Open Access, the authors have applied a CC BY public copyright license to any Author Accepted Manuscript version arising from this submission.

### Author contributions

Marta Matuszewska, Conceptualization, Data curation, Formal analysis, Investigation, Methodology, Writing – original draft, Writing – review and editing; Gemma GR Murray, Conceptualization, Formal analysis, Investigation, Methodology, Supervision, Visualization, Writing – original draft, Writing – review and editing; Xiaoliang Ba, Resources, Writing – review and editing; Rhiannon Wood, Resources; Mark A Holmes, Lucy A Weinert, Conceptualization, Funding acquisition, Supervision, Writing – review and editing

### Author ORCIDs

Marta Matuszewska ⓘ http://orcid.org/0000-0002-2653-7725
Gemma GR Murray ⓘ http://orcid.org/0000-0002-9531-1711

**Decision letter and Author response**

Decision letter https://doi.org/10.7554/eLife.74819.sa1

Author response https://doi.org/10.7554/eLife.74819.sa2

---

# Additional files

## Supplementary files

• Supplementary file 1. Strain names, country of origin, source (host species), year, accession numbers, and references for all isolates.

• Supplementary file 2. Genes that most strongly distinguish human- and livestock-associated CC398, and their association with mobile genetic elements. Our gene identifiers and the gene locations and locus tags in published reference genomes are provided.

• Supplementary file 3. Description of the presence of mobile genetic elements (MGEs) and annotation of MGE types and clades. The presence/absence of genes and MGEs is described by 1/0, and types and clades that are presented in the text are described.

• Supplementary file 4. Description of the genes in the Tn*916* element. The genes in the Tn*916* element used in our analyses are described in the reference genome 1_1439.

• Supplementary file 5. Reference SCC*mec* elements used in BLAST typing.

• Supplementary file 6. Description of the genes in the type V SCC*mec* element. The genes in the SCC*mec* type Vc element used in our analyses are described in the reference genome 12_LA_293.

• Supplementary file 7. AMR genes identified by PathogenWatch. Gene presence/absence is described by 1/0.

• Transparent reporting form

## Data availability

The Illumina sequences generated in this study have been deposited in the NCBI short read archive under the accession numbers ERR3524650, ERR3524328, ERR3524354, ERR3524446, and ERR3524562. All other sequences used in this study are publicly available and their origins are described in Supplementary file 1.

The following datasets were generated:

| Author(s) | Year | Dataset title | Dataset URL | Database and Identifier |
|---|---|---|---|---|
| Wellcome Sanger Institute | 2019 | Tracking_the_dynamics_ of_AMR_genes_within_ enteric_bacterial_ communities_in_pigs_and_ humans | https://www.ncbi. nlm.nih.gov/sra/ ERR3524650 | NCBI Sequence Read Archive, ERR3524650 |
| Wellcome Sanger Institute | 2019 | Tracking_the_dynamics_ of_AMR_genes_within_ enteric_bacterial_ communities_in_pigs_and_ humans | https://www.ncbi. nlm.nih.gov/sra/ ERR3524328 | NCBI Sequence Read Archive, ERR3524328 |
| Wellcome Sanger Institute | 2019 | Tracking_the_dynamics_ of_AMR_genes_within_ enteric_bacterial_ communities_in_pigs_and_ humans | https://www.ncbi. nlm.nih.gov/sra/ ERR3524354 | NCBI Sequence Read Archive, ERR3524354 |
| Wellcome Sanger Institute | 2019 | Tracking_the_dynamics_ of_AMR_genes_within_ enteric_bacterial_ communities_in_pigs_and_ humans | https://www.ncbi. nlm.nih.gov/sra/ ERR3524446 | NCBI Sequence Read Archive, ERR3524446 |
| Wellcome Sanger Institute | 2019 | Tracking_the_dynamics_ of_AMR_genes_within_ enteric_bacterial_ communities_in_pigs_and_ humans | https://www.ncbi. nlm.nih.gov/sra/ ERR3524562 | NCBI Sequence Read Archive, ERR3524562 |

The following previously published datasets were used:

| Author(s) | Year | Dataset title | Dataset URL | Database and Identifier |
|---|---|---|---|---|
| Gonçalves da Silva A | 2017 | Emergence of CC398 MRSA in New Zealand | https://www.ncbi.nlm.nih.gov/bioproject/?term=PRJEB12552 | NCBI BioProject, PRJEB12552 |
| Fox et al. | 2017 | Detection and molecular characterisation of Livestock-Associated MRSA in raw meat on retail sale in North West England | https://www.ncbi.nlm.nih.gov/bioproject/?term=PRJEB18725 | NCBI BioProject, PRJEB18725 |
| He et al. | 2018 | *Staphylococcus aureus* CC398 resequencing data | https://www.ncbi.nlm.nih.gov/bioproject/?term=PRJNA347471 | NCBI BioProject, PRJNA347471 |
| Heikinheimo et al. | 2016 | Studying 3 strains of LA-MRSA from Finland with interesting deletion genotypes | https://www.ncbi.nlm.nih.gov/bioproject/?term=PRJEB14187 | NCBI BioProject, PRJEB14187 |
| Himsworth et al. | 2014 | MRSA from rats and humans isolated from the Downtown Eastside of Vancouver, BC, Canada | https://www.ncbi.nlm.nih.gov/bioproject/?term=PRJEB5042 | NCBI BioProject, PRJEB5042 |
| Islam et al. | 2017 | Prevalence and origin of LA-MRSA CC398 in Danish horses | https://www.ncbi.nlm.nih.gov/bioproject/?term=PRJEB19362 | NCBI BioProject, PRJEB19362 |
| Larsen et al. | 2016 | *Staphylococcus aureus* | https://www.ncbi.nlm.nih.gov/bioproject/?term=PRJNA226567 | NCBI BioProject, PRJNA226567 |
| Lowder and Fitzgerald | 2018 | poultry-associated genetic elements in *Staphylococcus aureus* | https://www.ncbi.nlm.nih.gov/bioproject/?term=PRJNA312437 | NCBI BioProject, PRJNA312437 |
| Makarova et al. | 2017 | *Staphylococcus aureus* strain 08S00974 chromosome, complete genome | https://www.ncbi.nlm.nih.gov/nuccore/PRJNA378150 | NCBI GenBank, PRJNA378150 |
| Moller et al. | 2019 | Unusual MRSA CC398 hospital outbreak | https://www.ncbi.nlm.nih.gov/bioproject/?term=PRJNA508272 | NCBI BioProject, PRJNA508272 |
| Paterson et al. | 2013 | Diversity_of_MRSA | https://www.ncbi.nlm.nih.gov/bioproject/?term=PRJEB2655 | NCBI BioProject, PRJEB2655 |
| Holmes | 2018 | Tracking_the_dynamics_of_AMR_genes_within_enteric_bacterial_communities_in_pigs_and_humans | https://www.ncbi.nlm.nih.gov/bioproject/?term=PRJEB21015 | NCBI BioProject, PRJEB21015 |
| Price et al. | 2012 | *Staphylococcus aureus* CC398: host adaptation and emergence of methicillin resistance in livestock | https://www.ncbi.nlm.nih.gov/bioproject/?term=PRJNA274898 | NCBI BioProject, PRJNA274898 |
| Ronco et al. | 2018 | *Staphylococcus aureus* Raw sequence reads | https://www.ncbi.nlm.nih.gov/bioproject/?term=PRJNA430150 | NCBI BioProject, PRJNA430150 |
| Sharma et al. | 2016 | LA-MRSA CC398 from UK animals | https://www.ncbi.nlm.nih.gov/bioproject/?term=PRJEB14251 | NCBI BioProject, PRJEB14251 |

*Continued on next page*

*Continued*

| Author(s) | Year | Dataset title | Dataset URL | Database and Identifier |
|---|---|---|---|---|
| Sieber et al. | 2019 | Spread of LA-MRSA CC398 in pigs and humans in Denmark | https://www.ncbi.nlm.nih.gov/bioproject/?term=PRJEB25608 | NCBI BioProject, PRJEB25608 |
| Uhlemann et al. | 2017 | Sequencing of *Staphylococcus aureus* CC398 from Northern Manhattan | https://www.ncbi.nlm.nih.gov/bioproject/?term=PRJEB12818 | NCBI BioProject, PRJEB12818 |
| Ward et al. | 2014 | Global transmission and antibiotic resistance dynamics of *Staphylococcus aureus* CC398 in humans and livestock revealed by whole genome sequence analysis | https://www.ncbi.nlm.nih.gov/bioproject/?term=PRJEB7209 | NCBI BioProject, PRJEB7209 |
| Warne et al. | 2016 | *Staphylococcus_aureus__MSSA__study* | https://www.ncbi.nlm.nih.gov/bioproject/?term=PRJEB2755 | NCBI BioProject, PRJEB2755 |
| Zou et al. | 2021 | The *Staphylocossus aureus* isolates from central China | https://www.ncbi.nlm.nih.gov/bioproject/?term=PRJNA660925 | NCBI BioProject, PRJNA660925 |

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
