## [Editor Report]

The reviewers recognised the importance of understanding where new strains of microbes come from and how they change over time for infection control and prevention. *Staphylococcus aureus* CC398 is an important strain that 'spills over' from livestock to humans, carrying with it high levels of resistance to antibiotics commonly used in farming. This paper compares more than 1000 genomes of CC398 and concludes that spillover is likely to carry resistance to tetracyclines and other antibiotics into humans that will persist over time.

---

## [Decision Letter]

**Decision letter after peer review:**

Thank you for submitting your article "Human readaptation outpaces loss of antibiotic resistance in livestock-associated MRSA" for consideration by *eLife*. Your article has been reviewed by 3 peer reviewers, including Daniel J Wilson as Reviewing Editor and Reviewer #1, and the evaluation has been overseen by Gisela Storz as the Senior Editor.

Essential revisions:

The reviewers were all impressed by the excellent quality of these comprehensive analyses of an important pathogen strain. Please read the summary below in conjunction with the detailed reviews.

1) Our first main reservation was that the text should distinguish better what is new vs what is known throughout the paper. As an example, line 28-29 of the abstract is particularly ambiguous.

2) The second main reservation was the headline conclusion in the title -- that human re-adaptation (acquisition of human immune evasion genes by livestock-associated CC398s) outpaces loss of antibiotic resistance -- can be argued on first principles. From that perspective, it would appear to lack novelty.

3) A third point of discussion was an apples-and-oranges argument about comparing different MGE types. We felt that since they are fundamentally different and known to come and go at different rates, that this should be acknowledged with references. That is not to say there is no value to empirically quantifying those differences on timescales relevant to the evolution of CC398, and to discussing the implications for the turnover of the different AMR genes they carry.

My inclination is that the new analyses of MGE dynamics (how often known CC398-associated MGEs have come and gone, and the dates of these events) probably add sufficient novelty to warrant publication in *eLife*, but as currently presented that is not certain. I would welcome a revision that focuses on addressing this point.

*Reviewer #2 (Recommendations for the authors):*

I find that the comparison made in the title potentially a false equivalence between the evolutionary mechanisms driving gain/loss of pathogenicity genes, and gain/loss of resistance, for a few reasons. One, loss of resistance genes first requires that the 'mutation' leading to loss of resistance occurs, then lineages with that loss are selectively favoured. In contrast, (re)acquisition of the pathogenicity genes, which are already present in the wider population/community of bacteria, is much more likely to occur and then be selectively favoured. It is thus hard to envision a scenario where adaptive evolution for pathogenicity would ever be expected to outpace adaptive loss of resistance, unless the fitness costs of resistance are extremely high. I think the analysis and data stand on their own without having to force this comparison

Two, for an expectation that a 'thing' might be lost, its carriage should be associated with a fitness cost. While this is touched on, I would like to see more discussion about whether Tn916 is likely to carry a cost, otherwise its stable maintenance is the rule rather than the exception. Also, are there estimates about how easily Tn916 can be lost? Does it not drive its own reacquisition if an excision event is to occur?

Three, prophages, plasmids and transposons likely have different rates of gain/loss. This has been experimentally demonstrated by an excellent study from the group of Prof Jodi Lindsey (McCarthy et al., 2014, Genome Biology and Evolution, doi: 10.1093/gbe/evu214). Transfer of some MGEs was observed within 4 hours, whereas transfer of Tn916 was not observed within 16 days. I think it may lead to misleading comparisons to lump different types of MGEs together, as their gain/loss dynamics can differ significantly.

I would thus like to see an expanded discussion (i.e. more than just supposition) about estimated costs of SCCmec and phiSa3, or estimated benefits of (re-)acquiring these when selectively advantageous, and also what might be driving the replacement of Type V with other types. Also on relative rates of gain/loss events for the different categories of MGEs here. I would also recommend altering the title, as I do not find the comparison made there compelling, but that might be excessive.

*Reviewer #3 (Recommendations for the authors):*

In general, this manuscript is very well done and as mentioned in the public review, of high quality for the reader. It is in my eyes very close to be ready for publication if accepted, and I only have a few comments. Important, however, is the fact that the novelty in the manuscript is very limited because most results have been published earlier in the studies on the original datasets which are used here.

I only have a few comments:

General comments:

– Do you have any hypothesis on why ΦSa3 phages frequently are re-introduced and where from, as compared to other elements?

Specific comments:

– Line 62 and Table S1: Sieber et al., 2019 should be Sieber et al., 2018, mBio.

– Line 110: space missing before "(Ward et al., 2014)”.

– Line 275: “that integrate into the hlb gene of *S. aureus*”: There is strong evidence that the ΦSa3 phages integrate at other positions in LA-MRSA CC398. See e.g. Leinweber et al., 2021 (mBio) for a recent publication.

– Line 276: change "88% of our human-associated CC398 isolates" to "88% of the human-associated isolates in our collection".

– Lines 356-359: These results differ from what Sieber et al., 2019 found. Can you discuss this?

– Lines 416-418: This should be explained more precisely. Why are these dynamics consistent with these traits?

---

## [Author Response]

Essential revisions:The reviewers were all impressed by the excellent quality of these comprehensive analyses of an important pathogen strain. Please read the summary below in conjunction with the detailed reviews.

We would like to thank the editor and the reviewers for their positive assessments of our manuscript and useful suggestions. We have provided point-by-point responses to their comments below. The line numbers refer to the track-changes word document. In particular, we have made substantial additions to our discussion of our results, which we think has made our presentation and interpretation of our results clearer.

1) Our first main reservation was that the text should distinguish better what is new vs what is known throughout the paper. As an example, line 28-29 of the abstract is particularly ambiguous.

We agree that we could have made clearer which aspects of our results are novel, particularly in the abstract. While our study represents the largest comparative genomic study of CC398 isolates to date, allowing for more accurate characterisation of MGE carriage and dating of transitions, the greatest novelty of our study lies in our reconstruction of the evolutionary dynamics of the MGEs associated with the transition to livestock. We have now amended the manuscript in several places (including the abstract) to make this clearer through better referencing of previous work on CC398 and these MGEs (lines 26-30, 100-105, 199-201, 448-453, 482-500, 515-518, 522-533, 598-601).

2) The second main reservation was the headline conclusion in the title -- that human re-adaptation (acquisition of human immune evasion genes by livestock-associated CC398s) outpaces loss of antibiotic resistance -- can be argued on first principles. From that perspective, it would appear to lack novelty.

We agree that our interpretation of our results could have been more nuanced and have included more consideration of the mechanisms and selective pressures that may have led to the dynamics we identify. However, we don’t agree that our finding that adaptation to the human host occurs more rapidly than the loss of resistance can be argued from first principles.

In particular, MGEs carrying resistance genes often have fitness costs (e.g. Starikova et al., 2013 DOI:10.1093/jac/dkt270) (now noted in lines 527-529). Accurate predictions of the costs of MGEs are also difficult. This is in part because most studies of their impact on fitness are based on experimental (and often in vitro) assays, and these cannot fully replicate the selective pressures experienced by natural populations. Fitness costs can also vary across different bacterial hosts. In particular, the fitness costs of MGEs may be mitigated over time (lines 527-529), and therefore our finding that Tn*916* and SCC*mec* types V and IV have been maintained by livestock-associated CC398 over long periods may suggest that CC398 has evolved to mitigate costs associated with carriage of these MGEs.

A recent smaller-scale study (that didn’t fully correct for the impact of phylogenetic structure on their comparisons) came to the opposite conclusion to ours (Sieber et al., 2019; DOI:10.1038/s41598-019-55086-x) (lines 448-450). They concluded that transmission from pigs to humans is both associated with the loss of antibiotic resistance genes and the acquisition of human immune evasion genes. In reference to resistance genes, the authors concluded that ‘the genes are likely lost because they do not provide a fitness advantage in absence of the corresponding antimicrobial compounds in the human hosts’. The fact that this conclusion was drawn by authors who are experts in this field suggests that our findings are indeed novel.

In addition, the degree of selective benefit obtained from acquisition of the Sa3 prophage is not entirely clear. While these prophages are common in human *Staphylococcus aureus* strains and are associated with increased virulence, they are not essential for human nasal colonisation (Verkaik et al., 2011 DOI:10.1111/j.1469-0691.2010.03227.x).

Despite this, we agree that our title unnecessarily implied an equivalence in the evolutionary trajectories of the MGEs we are describing. We have therefore modified our title and added substantially to our discussion and interpretation of the dynamics we identify.

3) A third point of discussion was an apples-and-oranges argument about comparing different MGE types. We felt that since they are fundamentally different and known to come and go at different rates, that this should be acknowledged with references. That is not to say there is no value to empirically quantifying those differences on timescales relevant to the evolution of CC398, and to discussing the implications for the turnover of the different AMR genes they carry.My inclination is that the new analyses of MGE dynamics (how often known CC398-associated MGEs have come and gone, and the dates of these events) probably add sufficient novelty to warrant publication in eLife, but as currently presented that is not certain. I would welcome a revision that focuses on addressing this point.

We agree that our interpretation of our results should have included discussion of the different types of MGEs, as type influences short-term rates of acquisition and loss. As suggested, we have now added a paragraph to our discussion that directly addresses this point (lines 482-500), and greater context to our discussion of the dynamics of individual MGEs.

Nevertheless, we consider that there is considerable value in quantifying the long-term dynamics of these MGEs within CC398, independently of their type. Differences between these short-term and our long-term dynamics will reflect MGEs selective benefits/costs/availability. The long-term dynamics are important for understanding and predicting the evolution of the traits associated with the carriage of these MGEs. It is long-term dynamics that can inform predictions of how these traits are likely to be influenced by changes in the environment of CC398, such as host jumps or changes in the use of antibiotics in farming. Our current understanding of the long-term dynamics of MGEs, are how they are gained and lost in natural populations is extremely limited and we think our study develops a creative approach to uncovering these dynamics. Our amended discussion reflects on this point.

Reviewer #2 (Recommendations for the authors):I find that the comparison made in the title potentially a false equivalence between the evolutionary mechanisms driving gain/loss of pathogenicity genes, and gain/loss of resistance, for a few reasons. One, loss of resistance genes first requires that the 'mutation' leading to loss of resistance occurs, then lineages with that loss are selectively favoured. In contrast, (re)acquisition of the pathogenicity genes, which are already present in the wider population/community of bacteria, is much more likely to occur and then be selectively favoured. It is thus hard to envision a scenario where adaptive evolution for pathogenicity would ever be expected to outpace adaptive loss of resistance, unless the fitness costs of resistance are extremely high. I think the analysis and data stand on their own without having to force this comparison

We agree with the reviewer that our title could be interpreted as suggesting an equivalence in the processes and/or drivers of gain of pathogenicity genes and loss of resistance genes. As the reviewer notes, there are several factors that will influence the dynamics of these MGEs along a bacterial lineage, including those that influence the chance of random gain/loss of the MGE and the selective benefit/cost of this.

While we may expect (based on experiments, theory and observations of nature) that the selective benefit of losing MGEs carrying resistance genes in the absence of treatment is less than the benefit of acquiring MGEs carrying human-adaptive genes when moving into the human host, our results reveal how these predictions relate to the actual dynamics of these MGEs within CC398 as it moves between different host groups.

MGEs carrying resistance genes are often associated with fitness costs, and these costs can be host (and even insertion locus) specific (e.g. Starikova et al., 2013 DOI:10.1093/jac/dkt270) (now noted in lines 527-530). Accurate predictions of the costs of MGEs are made more difficult because most studies of their fitness costs are based on experimental (and often in vitro) assays, and these cannot fully replicate the selective pressures experienced by natural populations. It has also been suggested that the fitness costs of MGEs are likely to be mitigated over time (lines 527-530), and therefore our finding that particular MGEs have been maintained by livestock-associated CC398 over long periods may be informative because they predict that CC398 has evolved to mitigate the costs associated with carriage of these MGEs. Nevertheless, this cost could still vary across host species. For instance, it could be the case that CC398 experiences greater competition with resident populations of *Staphylococcus aureus* in a human host, and therefore that the cost carriage of MGEs carrying resistance genes may be greater in humans than in livestock.

Sieber *et al.,* (2019; DOI:10.1038/s41598-019-55086-x) compared the genomes of livestock-associated CC398 isolates sampled from human and pigs, and they concluded that transmission from pigs to humans is both associated with the loss of antibiotic resistance genes and the acquisition of human immune evasion genes. In reference to resistance genes, the authors concluded that ‘the genes are likely lost because they do not provide a fitness advantage in absence of the corresponding antimicrobial compounds in the human hosts’. This study had a smaller sample size (256 isolates) than ours, only included human and pig isolates, and compared human and pig isolates using a method that did not fully correct for phylogenetic structure. While our results are therefore not entirely comparable, we consider our results to be more robust than those of this previous study. However, the fact that this conclusion was drawn by authors who are experts in this field suggests that our findings are not entirely anticipated. We have added a reference to the results of this previous study to our manuscript (lines 448-450).

Nevertheless, we agree with the reviewer that the selection for loss of the MGEs carrying resistance genes in the absence of treatment is unlikely to be as strong as selection for acquisition of the MGEs carrying host-adaptive genes, and we have therefore modified our title so that we don’t imply a potentially false equivalence regarding the selective pressures on their carriage.

Two, for an expectation that a 'thing' might be lost, its carriage should be associated with a fitness cost. While this is touched on, I would like to see more discussion about whether Tn916 is likely to carry a cost, otherwise its stable maintenance is the rule rather than the exception. Also, are there estimates about how easily Tn916 can be lost? Does it not drive its own reacquisition if an excision event is to occur?

We have substantially expanded our discussion of the Tn*916* element (lines 502-533). In particular, we have added discussion about how its regulation may both mitigate its cost and promote its maintenance. While this may predict the stability of Tn*916*, this prediction does not hold for all bacterial species (such as *Streptococcus pnuemoniae*; D’Aeth et al., 2021) and therefore these mechanisms alone cannot explain or predict the maintenance of Tn*916* in CC398. We also discuss the likelihood that this element provides a strong selective benefit for CC398 in livestock due to the high levels of usage of tetracyclines, and how this element is observed across a wide range of bacterial species, including other opportunistic pathogens in the respiratory tract of pigs, and other *S. aureus* CCs. The latter suggests that the stability is not a consequence of the rarity of the element, as the element is likely to be present in other bacteria in the environment of CC398. Although it could instead reflect the rarity of successful interspecies transfers.

We have also added a section that describes the results of experimental studies of the dynamics of the Tn*916* transposon (and other MGEs) (lines 482-500). We report how these studies have found that transposons are more stably maintained (and less frequently transferred) than other types of MGEs. In particular, one study found that the Tn*916* in CC398 was stably inherited (and not transferred to other cells) over the course of a 16-day in vitro experiment. Nevertheless, the SCC*mec* and the φSa3 prophage were also stably maintained over the course of this experiment (while other prophages and plasmids were more dynamically gained and lost) (McCarthy et al., 2014). Therefore, while these short-term dynamics will likely influence the long-term dynamics we describe, other factors will also be important, and the long-term dynamics within a particular bacterial lineage cannot be predicted solely by transmission mechanism.

Three, prophages, plasmids and transposons likely have different rates of gain/loss. This has been experimentally demonstrated by an excellent study from the group of Prof Jodi Lindsey (McCarthy et al., 2014, Genome Biology and Evolution, doi: 10.1093/gbe/evu214). Transfer of some MGEs was observed within 4 hours, whereas transfer of Tn916 was not observed within 16 days. I think it may lead to misleading comparisons to lump different types of MGEs together, as their gain/loss dynamics can differ significantly.

We agree that in addition to differences in their selective benefit/cost different types of mobile genetic elements may differ in their dynamics due to differences in their mechanism of transfer. Nevertheless, our understanding of how these differences translate into long-term dynamics in natural populations is lacking in most cases, and studies such as ours can shed light on this. In the study undertaken by McCarthy et al., (2014), the Tn*916* element was not transferred during the 16 day experiment, but neither was the type V SCC*mec* or the Sa3 prophage. This suggests that all three of these elements are more intrinsically stable than other MGEs carried by CC398 isolates. This may, in part, explain the stability of all three of these elements in CC398 compared to other MGEs.

Other studies have, however, found that transposons tend to be more stable and less frequently transferred than prophages, and this difference could influence the long-term dynamics of these MGEs in CC398. These dynamics cannot, however, predict the long-term dynamics we identify. These long-term dynamics in a real population will be strongly influenced by both natural selection (both costs and benefits) and the availability of MGEs for acquisition by CC398. Therefore, despite what is known about short-term transmission dynamics of these MGEs, our study provides a unique insight into the longer-term dynamics these MGEs within CC398.

We agree with the reviewer that we ought to have related our results more to what is known of the dynamics of these elements over shorter time periods, and in laboratory conditions. We have therefore substantially expanded our discussion through the addition of a paragraph addressing this point (lines 482-500), and additions to our discussions of individual MGEs.

I would thus like to see an expanded discussion (i.e. more than just supposition) about estimated costs of SCCmec and phiSa3, or estimated benefits of (re-)acquiring these when selectively advantageous, and also what might be driving the replacement of Type V with other types. Also on relative rates of gain/loss events for the different categories of MGEs here. I would also recommend altering the title, as I do not find the comparison made there compelling, but that might be excessive.

We have provided a significantly expanded discussion which we hope addresses your concerns. We have tried to select the most relevant aspects of the literature on these three categories of elements to put our results in context, as there is a lot to discuss about each of these three elements. We have also amended our title, as you have suggested, so that we don’t suggest an equivalence between the drivers of the dynamics of these MGEs. We hope that we have satisfied you that the dynamics we describe are both interesting and important.

Reviewer #3 (Recommendations for the authors):In general, this manuscript is very well done and as mentioned in the public review, of high quality for the reader. It is in my eyes very close to be ready for publication if accepted, and I only have a few comments. Important, however, is the fact that the novelty in the manuscript is very limited because most results have been published earlier in the studies on the original datasets which are used here.

We would like to thank the reviewer for their positive assessment of our study and for their useful suggestions. We have provided a point-by-point response to their comments below. Page and line numbers refer to the word document with tracked changes.

We appreciate that we could have made it clearer which aspects of our results are novel. As our study is the largest comparative genomic analysis of CC398 isolates to date, it has allowed us to confirm previous described associations between MGEs and the transition of CC398 to livestock and allowed for more accurate dating of the emergence of CC398 in livestock. Nevertheless, the novelty of our study lies in our reconstruction of the evolutionary dynamics of the MGEs associated with the transition to livestock, which has never been done before. We have amended the manuscript in several places to make this clearer through better referencing of previous work on CC398 and these MGEs (lines 26-30, 100-105, 199-201, 448-453, 482-500, 515-518, 522-533, 598-601).

I only have a few comments:General comments:– Do you have any hypothesis on why ΦSa3 phages frequently are re-introduced and where from, as compared to other elements?

ΦSa3 prophages are common in human-associated *S. aureus* populations, and therefore it seems likely that when livestock-associated CC398 isolates are transmitted to humans they acquire them from other *S. aureus* CCs within the human host. The loss and reacquisition within human-associated CC398 could also result from occasional transmission between lineages that carry different ΦSa3 prophages when they co-colonise the same host. What is known about these elements suggests that they are only associated with *S. aureus*, and so their diversity and dynamics may reflect the large and diverse population of *S. aureus* that colonises human hosts (lines 598-601).

There is evidence that SCC*mec* has a larger host range, as it has been found more widely across staphylococci species. It is therefore possible that SCC*mec* acquisitions may originate from either other *S. aureus* lineages or from other staphylococci species. SCC*mec* type IV is associated with human community-associated *S. aureus* populations, and therefore might be acquired by livestock-associated CC398 when it moves through the human population (lines 571-578).

Tn*916* has an even larger host range. It has been found across several bacterial genera (lines 515-518). This makes it very difficult to determine where CC398 acquired its Tn*916* element from.

Specific comments:– Line 62 and Table S1: Sieber et al., 2019 should be Sieber et al., 2018, mBio.

Thank you for observing this error. We’ve now corrected this (line 83).

– Line 110: space missing before "(Ward et al., 2014)".

We’ve now corrected this (line 147).

– Line 275: "that integrate into the hlb gene of *S. aureus*": There is strong evidence that the ΦSa3 phages integrate at other positions in LA-MRSA CC398. See e.g. Leinweber et al., 2021 (mBio) for a recent publication.

Thank you for this correction. We have amended this to note that this is the primary integration site (line 343). In most, but not all, of our isolates carrying the Sa3 prophage we see evidence of insertion into the hlb gene.

– Line 276: change "88% of our human-associated CC398 isolates" to "88% of the human-associated isolates in our collection".

We’ve corrected this (lines 344-345).

– Lines 356-359: These results differ from what Sieber et al., 2019 found. Can you discuss this?

We have now mentioned this difference in our manuscript (lines 448-450). Our analysis differs from the one undertaken by Sieber et al., as it includes a larger number of isolates, isolates from livestock species other than pigs, and isolates from across a wider geographic range. In addition, due to the larger number of isolates in our collection, we were able to use an approach that better corrects for phylogenetic structure (comparing 70 phylogenetically independent pairs of isolates from human and livestock species). This means that our results are likely to be more robust than those of Sieber et al., 2019. While Sieber et al., 2019 considered patterns across three lineages (plus the remainder), they did not always observe consistent patterns of the loss of resistance genes across these independent lineages, and even when a consistent pattern was observed, the small number of independent comparisons means that these patterns may have arisen by chance.

– Lines 416-418: This should be explained more precisely. Why are these dynamics consistent with these traits?

We agree that this statement was unclear. We’ve now expanded on this statement, hopefully answering the reviewer’s question (lines 575-580). In short, the fact that the type V SCC*mec* has been replaced several times with type IV SCC*mec*, and that lineages carrying type IV SCC*mec* have persisted over several years (in parallel with those carrying type V), suggest that this replacement is not associated with a significant reduction in fitness. Multiple acquisitions of the type IV element and only a single acquisition of the type V element also suggests that the type IV element might be more common in the environments encountered by CC398.